

# Rapid Sea Ice Changes in the Future Barents Sea

Ole Rieke[1], Marius Årthun[1,2], and Jakob Simon Dörr[1,2]

[1]Geophysical Institute, University of Bergen, Norway
[2]Bjerknes Centre for Climate Research, Bergen, Norway

**Correspondence:** Marius Årthun (marius.arthun@uib.no)

**Abstract.** Winter Arctic sea ice loss is strongest in the Barents Sea. The anthropogenic ice decline is superimposed by pronounced internal variability that represents a large source of uncertainty in future climate projections. A notable manifestation of internal variability are periods of rapid ice loss or growth that greatly exceed the anthropogenic trend. These rapid ice change events are associated with large displacements of the sea ice edge which could potentially have both local and remote impacts on the climate system. In this study we accordingly present the first investigation of the frequency and drivers of these rapid ice change events in the future Barents Sea, using multi-member ensemble simulations from CMIP5 and CMIP6. A majority of rapid sea ice changes are triggered by trends in ocean heat transport or surface heat fluxes. Rapid ice change events are a common feature of the future Barents Sea until the region becomes close to ice free. As their evolution over time is closely tied to the average sea ice conditions, rapid ice changes in the Barents Sea serve as a precursor for future changes in adjacent seas.

## 1 Introduction

The Arctic is the region of most intense warming on the planet, with temperatures increasing twice as fast as the global average, i.e., an Arctic amplification of climate change (Cohen et al., 2020). The strong temperature increase is accompanied by a decline in sea ice thickness (Kwok, 2018) and extent (Onarheim et al., 2018) in all regions and all seasons. Future climate simulations project the strong sea ice decline to continue, leading to seasonally ice-free conditions in the Arctic as early as the middle of the 21st century (Notz and SIMIP Community, 2020; Årthun et al., 2021). However, future Arctic sea ice loss and the projected timing of ice-free conditions display a substantial spread across different models (Jahn et al., 2016). This large uncertainty results from model structure and emission scenario, but also internal climate variability (Swart et al., 2015; Bonan et al., 2021). Understanding internal variability is therefore important to predict future sea ice change under anthropogenic warming.

Whereas Arctic summer ice loss has largely occured in the central Arctic, winter ice loss has so far been confined to the outer shelf seas. The Barents Sea (Fig. 1) is the area of most intense winter sea ice area (SIA) loss, being on track towards year-round ice-free conditions in the second half of the 21st century (Onarheim and Årthun, 2017). A large part of the recent winter sea ice loss in the Barents Sea can be related to internal variability that is particularly strong in this region (England et al., 2019; Årthun et al., 2019; Bonan et al., 2021). On the sub-decadal timescale this variability is manifested in multi-year episodes of pronounced ice growth or ice loss that greatly exceed the long-term trend. These events of rapid ice changes are important to understand as they are characterized by substantial movements of the sea ice edge that have potential implications



for e.g., marine ecosystems (Fossheim et al., 2015; Sandø et al., 2021), shipping routes (Melia et al., 2016), and terrestrial climate (Lawrence et al., 2008; Zhang et al., 2018). Rapid ice loss events have been investigated for pan-Arctic summer sea ice (Holland et al., 2006; Auclair and Tremblay, 2018). And although there have been several studies conducted on interannual

winter sea ice variability in the Barents Sea (Kwok, 2009; Schlichtholz, 2011; Årthun et al., 2012; Nakanowatari et al., 2014; Skagseth et al., 2020), a detailed investigation of rapid ice change events is lacking.

In this study we accordingly present the first investigation of rapid ice change events (RICEs) in the Barents Sea using large ensemble climate model simulations. We first quantify the probability of rapid ice change events in present and future climates, demonstrating that strong – more than 7 times the observed ice decline – multi-year sea ice trends are a common

feature of the Barents Sea until it becomes close to ice-free, leading to substantial displacements of the sea ice edge on rather short timescales. The drivers of these rapid sea ice changes are thereafter investigated. Our analysis is largely based on a large ensemble simulation from the Community Earth System Model version 1, but the sensitivity of our results to model differences and future emission scenarios is also assessed using CMIP6 models.

## 2   Data and Methods

The main part of this analysis is based on future simulations from the Community Earth System Model Version 1 (CESM1; Hurrell et al., 2013), a fully coupled climate model that has a horizontal resolution of approximately 1° in all model components. We make use of two sets of experiments from the model. The large ensemble simulation (CESM-LE; Kay et al., 2015) consists of 40 members and covers the period from 1920-2100 based on historical greenhouse gas emissions until 2005 (Lamarque et al., 2010) and the RCP8.5 (Moss et al., 2010) thereafter. The other experiment applies an external greenhouse gas forcing

that limits global warming to 2°C (CESM-2C; Sanderson et al., 2017). This experiment consists of 11 members over the period 2006-2100. The model setup is identical to the CESM-LE with the external forcing as the only difference. The setups of the individual simulations differ only in slightly perturbed initial atmospheric conditions. All differences between the members are thus solely a result of internal variability (Deser et al., 2020). This allows us to split the variables into a common part (the ensemble mean) representing external forcing, and an individual part representing internal variability. In our analysis we

subtract the ensemble mean from each ensemble member to focus on internal variability.

As the Barents Sea is practically ice-free in summer, our analysis is based on winter means (November-April). To assess rapid ice changes we first calculate linear trends of Barents Sea ice area (15-60° E, 70-81° N; Fig. 1). We note that our results do not qualitatively change if we consider sea ice volume or sea ice extent instead. RICEs were then defined as linear trends that exceed a threshold of $7.7 * 10^4$ km$^2$yr$^{-1}$ over at least five years. This corresponds to two standard deviations of the distribution

of 5-year trends in CESM-LE between 2007 and 2025 (Fig. 2a), and is equivalent to 7 times the observational ice decline over the satellite era ($11.0*10^3$ km$^2$yr$^{-1}$ for 1979-2017, based on Walsh et al., 2017) Our results are not sensitive to the exact choice of this threshold. We apply the same threshold to CESM-2C and the CMIP6-models to enable direct comparison. Trends in ocean heat transport via the Barents Sea Opening (BSO), sea ice area transport between Franz Josef Land and Novaya Zemlya (eastern gateway) and between Svalbard and Franz Josef Land (northern gateway), net surface heat fluxes, sea level pressure





**Figure 1.** a) Observed winter (November-April) mean sea ice concentration (SIC; Walsh et al., 2017) and sea surface temperature (SST; Hersbach et al., 2019) in the Barents Sea (black box) between 2013 and 2017. Note the two different colorbars. The white line indicates the mean location of the winter ice edge (15%-SIC). b) Winter sea ice area in the Barents Sea from observations, the CESM-LE and CESM-2C. c) - f) Occurence of strong 5-year trends in SIC ($\geq$8%/yr) during different time periods of the CESM-LE simulations. The coloured lines indicate the southernmost (magenta) and northernmost (green) location of the ice edge during the respective time periods.

and surface air temperature have then been calculated over the duration of the events to assess their relationship to the RICEs. Ocean heat transport across Barents Sea Opening is calculated as

$$OHT_{BSO} = \int_S \rho c_p F dS \qquad (1)$$

with $\rho$ and $c_p$ the density and specific heat capacity of water, and $F$ the advective heat flux per unit volume (model variable UET). UET is calculated in the model using a reference temperature of 0°C. Sea ice area transport is calculated as the product of





**Table 1.** CMIP6 models used in the study

| Model | Ensemble Members | Reference |
|---|---|---|
| ACCESS-ESM1-5 | 10 | Ziehn et al. (2020) |
| CanESM5 | 10 | Swart et al. (2019) |
| EC-Earth3 | 15 | Döscher et al. (2021) |
| MIROC6 | 20 | Tatebe et al. (2019) |
| MPI-ESM1-2-LR | 10 | Mauritsen et al. (2019) |

sea ice concentration and ice drift velocity, integrated over the eastern and northern gateways. Sections are defined in alignment

with the native grid of the model. The results are not sensitive to their exact definition.

The CESM-LE has been used in several previous studies to investigate Arctic sea ice conditions and has been found to perform well (Auclair and Tremblay, 2018; Labe et al., 2018; England et al., 2019; Årthun et al., 2019; Dörr et al., 2021). The model slightly overestimates the sea ice cover in the Barents Sea as a result of lower simulated ocean temperatures than observed (Park et al., 2014). However, the observations (Walsh et al., 2017) fall within the ensemble spread (Fig. 1b). The

70 sensitivity of simulated Barents Sea ice extent to interannual variations in BSO heat transport is furthermore consistent with observations (Årthun et al., 2019). The model simulates ice import from the Kara Sea via the eastern gateway and variable ice transport across the northern gateway (not shown), both of which is in good agreement with observations (Lind et al., 2018).

To test the robustness of our results, we additionaly investigate RICEs in five CMIP6 climate models that have 10 or more ensemble members (Table 1), using both a high (SSP585) and a low (SSP126) warming scenario (O'Neill et al., 2017).

## 3 Barents Sea Ice Loss and Variability

Observed winter SIA in the Barents Sea has experienced an accelerating decline in the late 20th and early 21st century, resulting in a minimum SIA in 2017 which was only half of the 20th century mean (Fig. 1b). Future simulations under the assumption of the RCP8.5 climate scenario project a continuation of this decline towards an entirely ice-free Barents Sea by the end of this century (Fig. 1b; Onarheim and Årthun, 2017). The observed ice decline is overlaid by large interannual to decadal fluctuations,

indicative of strong internal variability. In the CESM simulations, this internal variability is expressed as an ensemble spread in SIA of approximately $\pm 30\%$. The magnitude of the internal variability remains quite constant over time until SIA becomes very low in CESM-LE (Fig. 1b). In CESM-2C, where SIA stabilises after 2050, the ensemble spread remains unchanged. The strength of internal variability can clearly be seen in the location of the southernmost and northernmost ice edges across the different ensemble members in CESM-LE (Fig. 1c-f). Although both shift northwards during the simulation, they encompass

a large area of possible locations. For example, for 2076-2100 (Fig. 1f) the ensemble spread includes an ice edge close to its present location but also one that has retreated past the boundaries of the Barents Sea.





**Figure 2.** Histograms of internally-driven 5-year trends of SIA (deviations from the ensemble mean) during different time periods in the 21st century for the CESM-LE and CESM-2C. The black solid line indicates the maximum 30-year ensemble mean ice decline in CESM-LE ($10.0*10^4 \mathrm{km}^2\mathrm{yr}^{-1}$ for 2031-2060), the dashed lines indicate the threshold for RICEs. The sample size (number of trends) of the histograms is indicated in the top-right corner. A 4th-order polynom has been removed from observations (Walsh et al., 2017) prior to calculating trends to represent the externally-forced signal (following Bonan et al., 2021).

## 4 Rapid Sea Ice Changes in CESM1

To quantify the occurrence of rapid ice change events in the Barents Sea, distributions of 5-year SIA-trends are presented in Fig. 2 for the CESM-LE and CESM-2C for different periods. The distribution of observed trends after 1920 is shown for comparison, and is similar to both, the historical simulations of CESM-LE (not shown) and the future simulations between 2007 and 2025 (Fig. 2a). Until 2050 CESM-LE and CESM-2C show similar distributions with many trends being much stronger



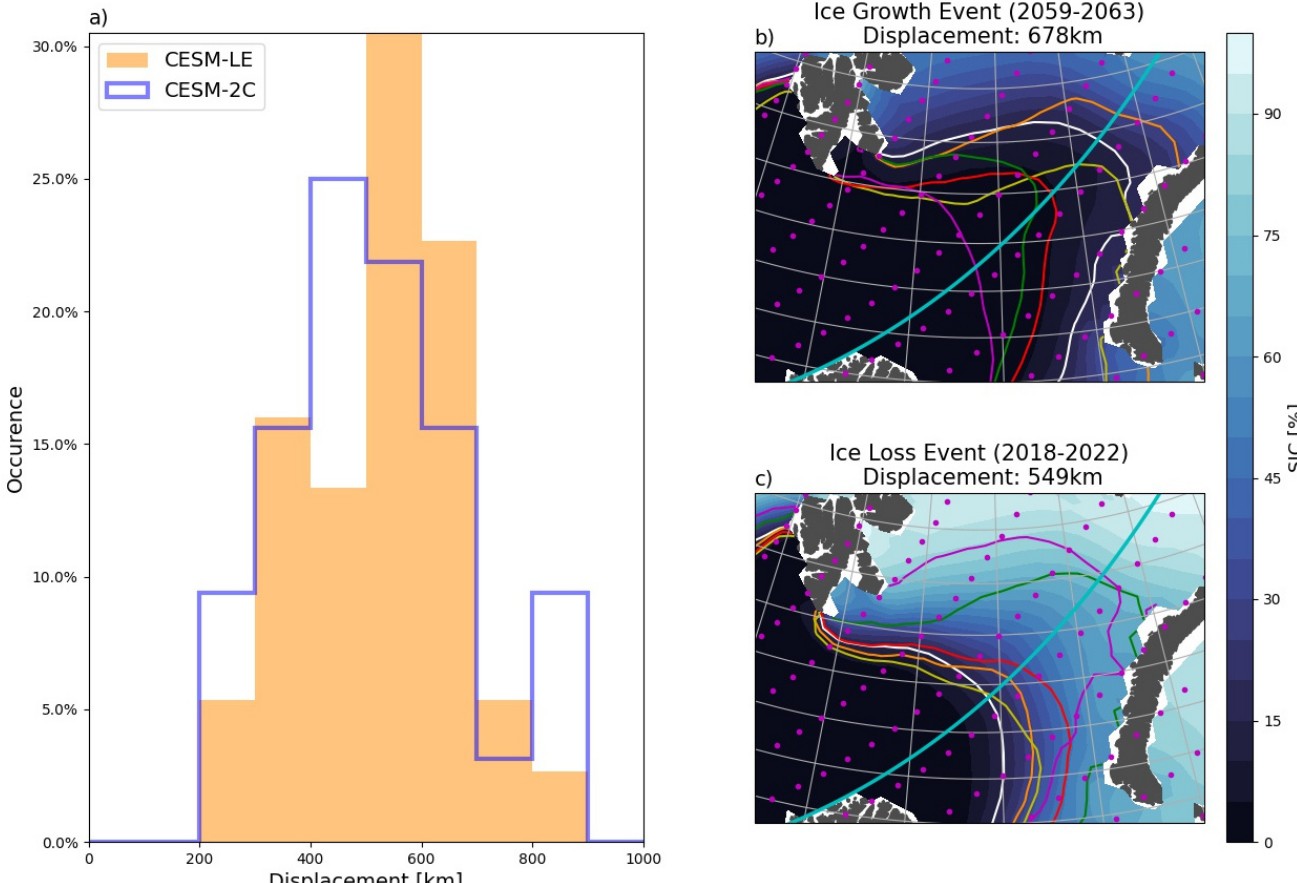

**Figure 3.** a) Distribution of ice edge displacement during RICEs in CESM-LE and CESM-2C. The displacement is calculated along the cyan line in panels b and c. b), c) Ice edge evolution during an example ice growth and ice loss event. Shading indicates the ensemble mean SIC during the respective time frame and the white line the ensemble mean ice edge (15%-SIC). The coloured lines indicate the ice edge (15%-SIC) during the RICEs in the order orange (first year), yellow, red, green, magenta (last year).

than the anthropogenic (ensemble mean) ice decline (indicated by the solid black line). In CESM-2C, the distributions in the second half of the 21st century remain quite similar to the previous time periods, as the average SIA remains rather constant during this time (Fig. 1b). In CESM-LE, however, the distribution becomes more confined towards smaller trends between

2051 and 2075, and even more so between 2076-2100, as the Barents Sea approaches ice-free conditions.

The spatial distribution of strong 5-year trends in CESM-LE ($\geq 8\%$ SIC/yr; the patterns are not sensitive to the exact choice of this threshold) is shown in Fig. 1c-f. It is seen that the area where large trends occur is shifting towards the boundaries of the Barents Sea in the north and the east. By the end of the century (2076-2100) this area has moved out of the boundaries of the Barents Sea as it is defined in this study, which results in the absence of strong SIA trends in the Barents Sea during that

time (Fig. 2d).





In the following, we will focus on the tails of the distributions, i.e., RICEs as these trends lead to the strongest changes in Barents Sea ice conditions. In CESM-LE we find 31 ice growth and 44 ice loss events between 2006 and 2100 and in CESM-2C we find 13 ice growth and 19 ice loss events that exceed our definition of a rapid ice change event (Section 2; vertical dashed lines in Fig. 2). This corresponds to an average of 2 RICEs per ensemble member in CESM-LE and 3 in CESM-2C.

105 The RICEs are associated with a large displacement of the ice edge, with ice loss (growth) events leading to a northward (southward) movement of the ice edge of up to 900 kilometres (Fig. 3). Ice loss events are on average related to a somewhat larger displacement than ice growth events, and the displacement also increases over time as sea ice retreats into the northeastern part of the Barents Sea (see the numbers given in Fig. 4). Two example cases from CESM-LE are depicted in Fig. 3. During an ice growth event in the second half of the 21st century, the ice edge is pushed 678 km southwards from close-to-average

sea ice conditions into the south-western part of the Barents Sea resulting in a present-day location (Fig. 3b). The example ice loss event in the early 21st century results in a rapid northward retreat of the ice edge (Fig. 3c). These examples emphasize the severity of RICEs as they can initiate a shift from average ice conditions to an unusually northward or southward location of the ice edge in only a few years. All ice growth events in CESM-LE, even those after 2050, result in an ice edge location very close to or even south of the present-day average (represented by the ensemble mean 2007-2025).

To understand and possibly predict these RICEs and their impacts, it is essential to identify the underlying mechanisms. We thus calculate the corresponding trend for potential drivers during each event, and assume those whose trend exceeds the threshold of one standard deviation to be related to that event. The relative importance of the investigated drivers is not sensitive to the exact choice of this threshold. Also note that as RICEs can be related to anomalous trends in more than one driver, ratios can add up to more than 100%. There are no significant differences between ice growth and ice loss events, and the forcing is

therefore evaluated for ice growth and ice loss events combined. Based on previous literature we consider three main drivers:

*Ocean heat transport:* Previous studies found a strong influence of BSO ocean heat transport on sea ice variability, with stronger (weaker) heat import leading to less (more) sea ice (Schlichtholz, 2011; Årthun et al., 2012; Docquier et al., 2021). We find ocean heat transport to be the most dominant driver of rapid ice changes. 79% (84%) of all RICEs in CESM-LE (CESM-2C) exhibit a simultaneous trend in OHT that exceeds one standard deviation (Fig. 4). For 5-year-trends the standard

deviation in OHT is 5.8 (5.2) $\mathrm{TW\,yr^{-1}}$. In comparison, the OHT increase needed to induce the observed sea ice loss in the Barents Sea since 1979 is approximately 1 $\mathrm{TW\,yr^{-1}}$ (Li et al., 2017).

*Sea ice transport*: An increase (decrease) in ice import can be associated with a growing (decreasing) sea ice cover, both via direct import and influences on local ice formation via stratification changes (Kwok, 2009; Lind et al., 2018). This is the case in 33% and 32% (22% and 25%) of the events for the northern and eastern gateway in CESM-LE (CESM-2C) (Fig. 4).

The threshold of one standard deviation is $7.4*10^4$ ($8.8*10^4$) $\mathrm{km^2yr^{-2}}$ for the northern and $4.3*10^4$ ($5.2*10^4$) $\mathrm{km^2yr^{-2}}$ for the eastern gateway. This would mean that even the strong increase in ice import between Svalbard and Franz-Josef-Land that observations show between 1999 and 2003 ($6.5*10^4$ $\mathrm{km^2yr^{-2}}$; Kwok, 2009) would be slightly too small to be considered relevant for triggering a RICE.

*Surface heat fluxes:* Changes in atmospheric circulation and associated heat and moisture transport can also influence the

135 sea ice cover (Woods and Caballero, 2016; Boisvert et al., 2016). In support of this, our results show a negative trend in sea



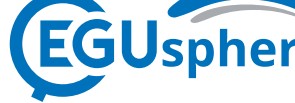

**Figure 4.** The fraction of RICEs that show a simultaneous trend in the respective forcing parameter of more than one standard deviation. The drivers are BSO ocean heat transport (OHT), ice transport through the northern (ITN) and eastern gateway (ITE) and surface heat flux (SHF) in the southwestern Barents Sea (Figure 5c).

level pressure over the Fram Strait during ice loss events (Fig. 5a), associated with strengthening westerly winds over BSO and southerly winds over the central and northern Barents Sea. As a result, surface air temperatures increase in the northern Barents Sea (Fig. 5b) during ice loss. Warmer westerly winds also lead to reduced ocean heat loss in the ice-free southern Barents Sea, whereas more heat is lost in the northern Barents Sea as a result of more open ocean area (Fig. 5c; Skagseth

et al., 2020). Considering surface heat fluxes in the permanently ice-free southwestern Barents Sea (16-38° E; 71-76° N) as a fingerprint of atmospheric forcing of ocean temperature and, hence, sea ice (Schlichtholz and Houssais, 2011), we find 65% (62%) of the RICEs to be associated with anomalous trends in SHF in CESM-LE (CESM-2C); decreasing (increasing) ocean heat loss corresponding to SIA decline (increase).

 

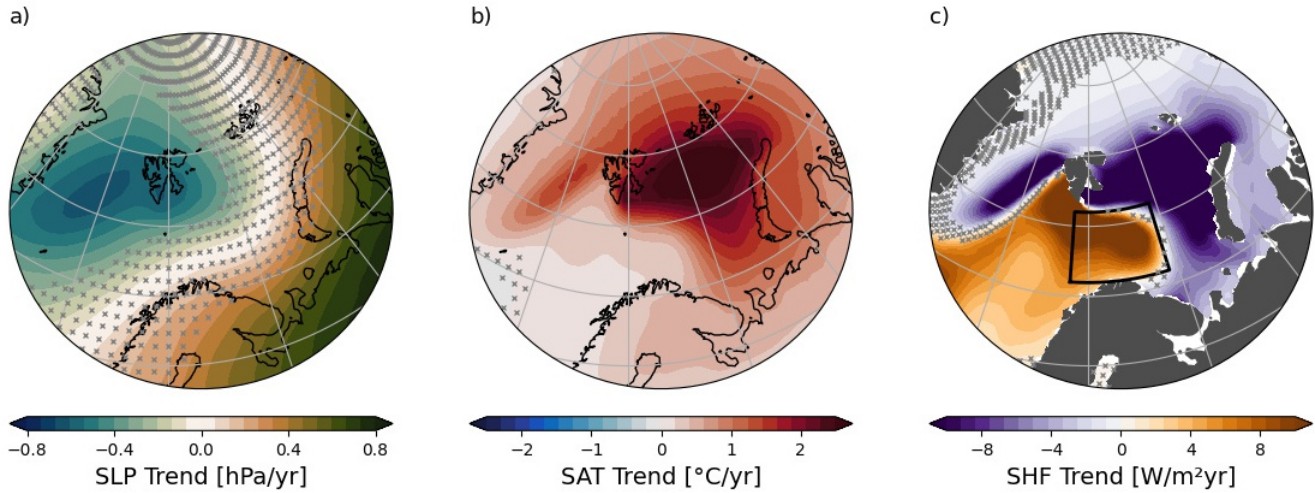

**Figure 5.** Linear trend of a) sea level pressure (SLP), b) surface air temperature (SAT) and c) surface heat flux (SHF) anomalies during ice loss events, averaged over all events. Ocean heat loss is defined as negative, meaning that positive (negative) anomalies refer to less (more) heat loss. The black box indicates the area for averaging SHF to assess its influence on RICEs. Crosses indicate areas where the trend is not statistically significant at the 95% confidence level.

Although the different drivers have been assessed and quantified individually, they are to some extent interconnected. For example, ice loss events are associated with an anomalous atmospheric circulation (Fig. 5a) that will influence both ocean heat transport (Herbaut et al., 2015), surface heat fluxes (Skagseth et al., 2020), and sea ice area transport (Kwok, 2009). 51 (50) % of the RICEs exhibit significant trends in both, OHT and SHF, emphasiszing their interconnection. A detailed analysis of these relationships is not presented here. However, removing (by regression) the linear signal associated with OHT from time series of regional winds over the Barents Sea suggests that atmospheric circulation (wind) anomalies are mainly effecting the sea ice cover through changes in BSO heat transport, consistent with the findings of e.g. Lien et al. (2017).

In general, we find that the relative importance of the different drivers varies somewhat during the simulations, but no systematic change is seen (Fig. 4). Differences between CESM-LE and CESM-2C are also small and show no consistency, suggesting that the forcing of RICEs remains unaffected by the underlying ice conditions.

## 5 Rapid Sea Ice Changes in CMIP6 models

The occurrence of rapid sea ice changes is further studied in a suite of CMIP6 models. Figure 6 shows the distribution of 5-year trends from the CMIP6 models under two emission scenarios. Most models show a distribution of trends that is fairly similar to, yet slightly narrower than the CESM. An exception is CanESM5 that simulates much weaker trends than the other models, likely as a result of the low average SIA in the model after the strong ice loss until 2030 (Fig. 6). During the 21st century SIA decreases in all model simulations and the distributions become more confined to weaker trends. Although the models agree on





**Figure 6.** Violin plots showing the distribution of 5-year trends of SIA during different episodes of the 21st century in different model simulations. The orange and blue vertical lines indicate the 95-percentile for SSP585 and SSP126, respectively. The black vertical lines indicate the strongest 30-year ice decline in the ensemble mean of the respective SSP585 simulation. The numbers indicate the average SIC in the Barents Sea in the SSP585 (left) and SSP126 (right) experiment for each model during the respective time periods. The bottom panels show the ensemble mean SIA in the different simulations. For CESM the colors indicate the CESM-LE (RCP8.5; orange) and the CESM-2C (blue).

this general behaviour, the future changes in trends differ as a result of the different rates of SIA decline in each model. Trends simulated in the SSP126 experiments are generally stronger compared to the SSP585 experiments after 2050, consistent with





a larger SIA. Only the MPI-ESM1-2-LR simulates a stabilisation of the SIA in the Barents Sea under a low-emission scenario (in agreement with CESM-2C), whereas the other CMIP6 models show practically ice-free conditions at the end of the 21st century even under SSP126 (see also Årthun et al., 2021).

The different mean states in the models are also reflected in the number of RICEs. The CMIP6 model that simulates the largest average SIC in the Barents Sea, EC-Earth3, also simulates most RICEs per ensemble member (1.6/2.2 in the SSP585/SSP126-simulation) using the same criterion of $7.7 * 10^4 \text{km}^2\text{yr}^{-1}$ (7 times the observational ice loss). However, this model exhibits a very strong anthropogenic ice decline (represented by the maximum 30-year ensemble mean SIA trend of -15.2*$10^3\text{km}^2\text{yr}^{-1}$ in SSP585 between 1993 and 2022; Fig. 6j), which leads to RICEs in EC-Earth3 being weaker relative to the

long-term ice loss than in CESM. In contrast, RICEs in ACCESS ESM1 (0.4/1.7 in SSP585/SSP126) are much stronger than the anthropogenic ice loss which is rather small (-7.4*$10^3\text{km}^2\text{yr}^{-1}$ between 1981 and 2010 under SSP585). RICEs can also be found in MPI-ESM (0.4/0.5) and MIROC6 (0.2/0.2), that simulate average ice loss similar to the CESM-LE (-8.3 (1987-2016) and -9.8*$10^3\text{km}^2\text{yr}^{-1}$ (2016-2045) in SSP585). Only CanESM simulates no RICEs whatsoever in either simulation. This model is characterised by a very strong anthropogenic ice loss (-18.8*$10^3\text{km}^2\text{yr}^{-1}$ between 1993 and 2022), resulting in

ice-free conditions as early as 2025. CanESM5 is also the model with the weakest internal variability, evident from the very narrow distribution of sea ice trends (Fig. 6c; also quantified by the ensemble spread in SIA). The weak internal variability in this model has also been noted in other studies (Bonnet et al., 2021). In general, we find that models with stronger internal variability produce more RICEs. We thus conclude that although the CESM seems to represent an upper bound for RICEs in the Barents Sea, they generally occur also in other CMIP6 models. Model differences in the occurrence of RICEs are closely

related to average sea ice conditions and the strength of internal variability.

## 6   Discussion and Conclusion

The Barents Sea is the area of most intense winter sea ice loss and future projections show a continued decline towards ice-free conditions by the end of this century (Fig. 1 Onarheim and Årthun, 2017). Internal variability in the climate system leads to large interannual to decadal fluctuations superimposed on this long-term trend (England et al., 2019). A visible manifestation

of these internally-driven fluctuations is the occurrence of large, abrupt changes in the sea ice cover. These rapid ice change events (RICEs) are several times stronger than the externally-forced ice loss and can hence lead to an acceleration, pausing or reverse of the ice decline. In this study we present the first investigation of RICEs in the Barents Sea. We use outputs from two ensemble simulations from CESM and multi-member CMIP6 simulations to investigate the future evolution of winter sea ice variability in the Barents Sea under different emission scenarios. Although CESM simulates the largest number of RICEs,

possibly representing an upper bound for their occurence, RICEs are also found at similar rates in most other models. The occurence of RICEs is directly related to average sea ice conditions and hence to future emissions.

    Rapid ice loss events have previously been studied in future climate simulations for the pan-Arctic in summer. Holland et al. (2006) and Auclair and Tremblay (2018) find most of those pan-Arctic events to be associated with anomalies in ocean heat transport which is consistent with our results for the Barents Sea. In addition to OHT we also investigate the influence

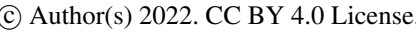



of other parameters and find a substantial contribution from surface heat fluxes and ice transport. This is largely consistent with the results from studies focusing on interannual variability in the Barents Sea (Kwok, 2009; Schlichtholz, 2011; Årthun et al., 2012; Nakanowatari et al., 2014; Skagseth et al., 2020). We emphasize that this is not a priori granted, and note that distinct mechanisms have been identified for interannual variability and long-term trends in BSO ocean heat transport (Wang et al., 2019). Venegas and Mysak (2000) also found different dominant mechanisms of sea ice variability in the Barents Sea for different time scales. We find no systematic change of the underlying drivers over time, between the emission scenarios or between ice growth and loss events. From this we infer that the underlying processes of driving rapid ice changes in the Barents Sea remain unaffected by global warming and the retreating sea ice.

In this study we have shown the importance of rapid ice changes in the Barents Sea. RICEs are especially important due to the substantial movements of the ice edge, which, as the border between ice-covered and open ocean, is of large importance for climate (e.g., Zhang et al., 2018) and ecosystem processes (e.g., Fossheim et al., 2015). Identifying the leading drivers of RICEs is therefore crucial for understanding and predicting such events and their associated broad impacts. When the Barents Sea approaches ice-free conditions, the area experiencing rapid sea ice changes will retreat past the boundaries of the Barents Sea into the central Arctic and the Kara Sea, a visible footprint of the future Atlantification of the Arctic Ocean (Fig. 1f; Dörr et al., 2021; Shu et al., 2021). Our results could therefore provide important insight into future sea ice variability in other Arctic seas.

*Data availability.* All data in this study are publicly available. Output from CESM is available via the Earth System Grid: https://www.earthsystemgrid.org. CMIP6 data are available from the Earth System Grid Federation (ESGF) (e.g., https://esgf-node.llnl.gov/search/cmip6). Observed Arctic SICs are available from https://nsidc.org/data/G10010. SST from ERA5 is available through the Copernicus Climate Change Service: https://cds.climate.copernicus.eu/cdsapp#!/dataset/10.24381/cds.f17050d7

*Author contributions.* MÅ conceived the study. OR performed the analysis, produced the figures, and wrote the paper. MÅ and JD contributed to improving the manuscript. All authors contributed to the methods design, results analysis and manuscript reviewing

*Competing interests.* The authors declare no conflict of interests

*Acknowledgements.* This study was funded by the Research Council of Norway project Nansen Legacy (Grant 276730), and the Trond Mohn Foundation (Grant BFS2018TMT01). We thank L.H. Smedsrud and A.B. Sandø for input and suggestions.



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
