# Peer review of "Rapid Sea Ice Changes in the Future Barents Sea"

_EGUsphere, 2022_

## Referee Comment (RC1)

**Rapid Sea Ice Changes in the Future Barents Sea**

Ole Rieke[1], Marius Årthun[1,2], and Jakob Simon Dörr[1,2]

[1]Geophysical Institute, University of Bergen, Norway
[2]Bjerknes Centre for Climate Research, Bergen, Norway

**Correspondence:** Marius Årthun (marius.arthun@uib.no)

**Abstract.** Winter Arctic sea ice loss is strongest in the Barents Sea. The anthropogenic ice decline is superimposed by pronounced internal variability that represents a large source of uncertainty in future climate projections. A notable manifestation of internal variability are periods of rapid ice loss or growth that greatly exceed the anthropogenic trend. These rapid ice change events are associated with large displacements of the sea ice edge which could potentially have both local and remote impacts on the climate system. In this study we accordingly present the first investigation of the frequency and drivers of these rapid ice change events in the future Barents Sea, using multi-member ensemble simulations from CMIP5 and CMIP6. A majority of rapid sea ice changes are triggered by trends in ocean heat transport or surface heat fluxes. Rapid ice change events are a common feature of the future Barents Sea until the region becomes close to ice free. As their evolution over time is closely tied to the average sea ice conditions, rapid ice changes in the Barents Sea serve as a precursor for future changes in adjacent seas.

**1   Introduction**

The Arctic is the region of most intense warming on the planet, with temperatures increasing twice as fast as the global average, i.e., an Arctic amplification of climate change (Cohen et al., 2020). The strong temperature increase is accompanied by a decline in sea ice thickness (Kwok, 2018) and extent (Onarheim et al., 2018) in all regions and all seasons. Future climate simulations project the strong sea ice decline to continue, leading to seasonally ice-free conditions in the Arctic as early as the middle of the 21st century (Notz and SIMIP Community, 2020; Årthun et al., 2021). However, future Arctic sea ice loss and the projected timing of ice-free conditions display a substantial spread across different models (Jahn et al., 2016). This large uncertainty results from model structure and emission scenario, but also internal climate variability (Swart et al., 2015; Bonan et al., 2021). Understanding internal variability is therefore important to predict future sea ice change under anthropogenic warming.

Whereas Arctic summer ice loss has largely occured in the central Arctic, winter ice loss has so far been confined to the outer shelf seas. The Barents Sea (Fig. 1) is the area of most intense winter sea ice area (SIA) loss, being on track towards year-round ice-free conditions in the second half of the 21st century (Onarheim and Årthun, 2017). A large part of the recent winter sea ice loss in the Barents Sea can be related to internal variability that is particularly strong in this region (England et al., 2019; Årthun et al., 2019; Bonan et al., 2021). On the sub-decadal timescale this variability is manifested in multi-year episodes of pronounced ice growth or ice loss that greatly exceed the long-term trend. These events of rapid ice changes are important to understand as they are characterized by substantial movements of the sea ice edge that have potential implications

**Number: 1 Author: anonymous Subject: Highlight Date: 11/07/2022, 16:00:32**

I have been asked to review the manuscript "Rapid Sea ICe CHanges in the Future Barents Sea" by Rieke et al.
The topic of the work is important for current climate resarch, as the Arctic is constantly changing alongside global warming. In particular, the Barents Sea is a part of the region representing an important connection between the Arctic basins and the remainder of the Eurasian shelves, and lower latitudes.

Whereas the data basis for analysis appears state-of-the-art and the analytical approach overall sound there are several details that I suggest be clarified before publication.

The method of relating different derived quantities to changes in sea-ice cover is not clearly explained. The calculation of heat transport has, in the past, been challenged, and at least a few sentences of explanation why the approach used here is valid would be in order.

Thus, I suggest to publish this manuscript, subject to the modest corrections outlined in my comments. Note that I submit the comments as PDF comments in the form of a comment summary.

**Number: 2 Author: anonymous Subject: Inserted Text Date: 11/07/2022, 15:25:38**

could also cite the current IPCC report: Meredith, M., M. Sommerkorn, S. Cassotta, C. Derksen, A. Ekaykin, A. Hollowed, G. Kofinas, A. Mackintosh, J. Melbourne-Thomas, M.M.C. Muelbert, G. Ottersen, H. Pritchard, and E.A.G. Schuur, 2019: Polar Regions. In: IPCC Special Report on the Ocean and Cryosphere in a Changing Climate [H.-O. Pörtner, D.C. Roberts, V. Masson-Delmotte, P. Zhai, M. Tignor, E. Poloczanska, K. Mintenbeck, A. Alegría, M. Nicolai, A. Okem, J. Petzold, B. Rama, N.M. Weyer (eds.)]. In press.)

**Number: 3 Author: anonymous Subject: Inserted Text Date: 11/07/2022, 15:23:23**

scenarios

for e.g., marine ecosystems (Fossheim et al., 2015; Sandø et al., 2021), shipping routes (Melia et al., 2016), and terrestrial climate (Lawrence et al., 2008; Zhang et al., 2018). Rapid ice loss events have been investigated for pan-Arctic summer sea ice (Holland et al., 2006; Auclair and Tremblay, 2018). And although there have been several studies conducted on interannual
30   winter sea ice variability in the Barents Sea (Kwok, 2009; Schlichtholz, 2011; Årthun et al., 2012; Nakanowatari et al., 2014; Skagseth et al., 2020), a detailed investigation of rapid ice change events is lacking.

In this study we accordingly present the first investigation of rapid ice change events (RICEs) in the Barents Sea using large ensemble climate model simulations. We first quantify the probability of rapid ice change events in present and future climates, demonstrating that strong – more than 7 times the observed ice decline – multi-year sea ice trends are a common
35   feature of the Barents Sea until it becomes close to ice-free, leading to substantial displacements of the sea ice edge on rather short timescales. The drivers of these rapid sea ice changes are thereafter investigated. Our analysis is largely based on a large ensemble simulation from the Community Earth System Model version 1, but the sensitivity of our results to model differences and future emission scenarios is also assessed using CMIP6 models.

**2   Data and Methods**

40   The main part of this analysis is based on future simulations from the Community Earth System Model Version 1 (CESM1; Hurrell et al., 2013), a fully coupled climate model that has a horizontal resolution of approximately 1° in all model components. We make use of two sets of experiments from the model. The large ensemble simulation (CESM-LE; Kay et al., 2015) consists of 40 members and covers the period from 1920-2100 based on historical greenhouse gas emissions until 2005 (Lamarque et al., 2010) and the RCP8.5 (Moss et al., 2010) thereafter. The other experiment applies an external greenhouse gas forcing
45   that limits global warming to 2°C (CESM-2C; Sanderson et al., 2017). This experiment consists of 11 members over the period 2006-2100. The model setup is identical to the CESM-LE with the external forcing as the only difference. The setups of the individual simulations differ only in slightly perturbed initial atmospheric conditions. All differences between the members are thus solely a result of internal variability (Deser et al., 2020). This allows us to split the variables into a common part (the ensemble mean) representing external forcing, and an individual part representing internal variability. In our analysis we
50   subtract the ensemble mean from each ensemble member to focus on internal variability.

As the Barents Sea is practically ice-free in summer, our analysis is based on winter means (November-April). To assess rapid ice changes we first calculate linear trends of Barents Sea ice area (15-60° E, 70-81° N; Fig. 1). We note that our results do not qualitatively change if we consider sea ice volume or sea ice extent instead. RICEs were then defined as linear trends that exceed a threshold of $7.7 * 10^4$ km$^2$yr$^{-1}$ over at least five years. This corresponds to two standard deviations of the distribution
55   of 5-year trends in CESM-LE between 2007 and 2025 (Fig. 2a), and is equivalent to 7 times the observational ice decline over the satellite era ($11.0*10^3$ km$^2$yr$^{-1}$ for 1979-2017, based on Walsh et al., 2017). Our results are not sensitive to the exact choice of this threshold. We apply the same threshold to CESM-2C and the CMIP6-models to enable direct comparison. Trends in ocean heat transport via the Barents Sea Opening (BSO), sea ice area transport between Franz Josef Land and Novaya Zemlya (eastern gateway) and between Svalbard and Franz Josef Land (northern gateway), net surface heat fluxes, sea level pressure

Number: 1 Author: anonymous   Subject: Highlight   Date: 11/07/2022, 15:27:02
Please specify what reange of choices you explored (e.g. one standard deviation, three standard deviations...).

Number: 2 Author: anonymous   Subject: Inserted Text   Date: 11/07/2022, 15:25:59
.

[Figure]

[Figure]

**Figure 1.** a) Observed winter (November-April) mean sea ice concentration (SIC; Walsh et al., 2017) and sea surface temperature (SST; Hersbach et al., 2019) in the Barents Sea (black box) between 2013 and 2017. Note the two different colorbars. The white line indicates the mean location of the winter ice edge (15%-SIC). b) Winter sea ice area in the Barents Sea from observations, the CESM-LE and CESM-2C. c) - f) Occurence of strong 5-year trends in SIC (≥8%/yr) during different time periods of the CESM-LE simulations. The coloured lines indicate the southernmost (magenta) and northernmost (green) location of the ice edge during the respective time periods.

60   and surface air temperature have then been calculated over the duration of the events to assess their relationship to the RICEs. Ocean heat transport across Barents Sea Opening is calculated as

$$OHT_{BSO} = \int_S \rho c_p F dS \qquad (1)$$

with $\rho$ and $c_p$ the density and specific heat capacity of water, and $F$ the advective heat flux per unit volume (model variable UET). UET is calculated in the model using a reference temperature of 0°C. Sea ice area transport is calculated as the product of

Number: 1 Author: anonymous   Subject: Inserted Text          Date: 11/07/2022, 15:27:26

(SIA)

Number: 2 Author: anonymous   Subject: Highlight      Date: 11/07/2022, 15:30:41

This appears to be an arbitrary choice. Seawater at typical Barents Sea salinities in winter freezes around -2 deg C. Please elaborate a bit more how sensitive your results are to this choice of reference temperature. Refer to / cite the following publication:

Schauer, U. and Beszczynska-Möller, A.: Problems with estimation and interpretation of oceanic heat transport – conceptual remarks for the case of Fram Strait in the Arctic Ocean, Ocean Sci., 5, 487–494, https://doi.org/10.5194/os-5-487-2009, 2009.

[Figure]

[Figure]

**Figure 2.** Histograms of internally-driven 5-year trends of SIA (deviations from the ensemble mean) during different time periods in the 21st century for the CESM-LE and CESM-2C. The black solid line indicates the maximum 30-year ensemble mean ice decline in CESM-LE ($10.0*10^4 km^2 yr^{-1}$ for 2031-2060), the dashed lines indicate the threshold for RICEs. The sample size (number of trends) of the histograms is indicated in the top-right corner. A 4th-order polynom has been removed from observations (Walsh et al., 2017) prior to calculating trends to represent the externally-forced signal (following Bonan et al., 2021).

**4 Rapid Sea Ice Changes in CESM1**

To quantify the occurrence of rapid ice change events in the Barents Sea, distributions of 5-year SIA-trends are presented in Fig. 2 for the CESM-LE and CESM-2C for different periods. The distribution of observed trends after 1920 is shown for comparison, and is similar to both, the historical simulations of CESM-LE (not shown) and the future simulations between 2007 and 2025 (Fig. 2a). Until 2050 CESM-LE and CESM-2C show similar distributions with many trends being much stronger

Number: 1 Author: anonymous   Subject: Highlight     Date: 11/07/2022, 15:32:54

I know this may seem picky, but if you used the same colours for the nLE, n2C and nObs as in the legend it would be intuitively easier to associate the "n" to the correct set of data.

[Figure]

In the following, we will focus on the tails of the distributions, i.e., RICEs as these trends lead to the strongest changes in Barents Sea ice conditions. In CESM-LE we find 31 ice growth and 44 ice loss events between 2006 and 2100 and in CESM-2C we find 13 ice growth and 19 ice loss events that exceed our definition of a rapid ice change event (Section 2; vertical dashed lines in Fig. 2). This corresponds to an average of 2 RICEs per ensemble member in CESM-LE and 3 in CESM-2C.

105 The RICEs are associated with a large displacement of the ice edge, with ice loss (growth) events leading to a northward (southward) movement of the ice edge of up to 900 kilometres (Fig. 3). Ice loss events are on average related to a somewhat larger displacement than ice growth events, and the [1]displacement also increases over time as sea ice retreats into the northeastern part of the Barents Sea (see the numbers given in Fig. 4). Two example cases from CESM-LE are depicted in Fig. 3. During an ice growth event in the second half of the 21st century, the ice edge is pushed 678 km southwards from close-to-average

110 sea ice conditions into the south-western part of the Barents Sea resulting in a present-day location (Fig. 3b). The example ice loss event in the early 21st century results in a rapid northward retreat of the ice edge (Fig. 3c). These examples emphasize the severity of RICEs as they can initiate a shift from average ice conditions to an unusually northward or southward location of the ice edge in only a few years. All ice growth events in CESM-LE, even those after 2050, result in an ice edge location very close to or even south of the present-day average (represented by the ensemble mean 2007-2025).

115 To understand and possibly predict these RICEs and their impacts, it is essential to identify the underlying mechanisms. [2]We thus calculate the corresponding trend for potential drivers during each event, and assume those whose trend exceeds the threshold of one standard deviation to be related to that event. The relative importance of the investigated drivers is not sensitive to the exact choice of this threshold. [3]Also note that as RICEs can be related to anomalous trends in more than one driver, ratios can add up to more than 100%. There are no significant differences between ice growth and ice loss events, and the forcing is

120 therefore evaluated for ice growth and ice loss events combined. Based on previous literature we consider three main drivers:

[4]*Ocean heat transport:* Previous studies found a strong influence of BSO ocean heat transport on sea ice variability, with stronger (weaker) heat import leading to less (more) sea ice (Schlichtholz, 2011; Årthun et al., 2012; Docquier et al., 2021). We find ocean heat transport to be the most dominant driver of rapid ice changes. 79% (84%) of all RICEs in CESM-LE (CESM-2C) exhibit a simultaneous trend in [5]OHT that exceeds one standard deviation (Fig. 4). For 5-year-trends the standard

125 deviation in OHT is 5.8 (5.2) TW yr$^{-1}$. In comparison, the OHT increase needed to induce the observed sea ice loss in the Barents Sea since 1979 is approximately 1 TW yr$^{-1}$ (Li et al., 2017).

*Sea ice transport*: An increase (decrease) in ice import can be associated with a growing (decreasing) sea ice cover, both via direct import and influences on local ice formation via stratification changes (Kwok, 2009; Lind et al., 2018). This is the case in 33% and 32% (22% and 25%) of the events for the northern and eastern gateway in CESM-LE (CESM-2C) (Fig. 4).

130 The threshold of one standard deviation is $7.4*10^4$ ($8.8*10^4$) km$^2$yr$^{-2}$ for the northern and $4.3*10^4$ ($5.2*10^4$) km$^2$yr$^{-2}$ for the eastern gateway. This would mean that even the strong increase in ice import between Svalbard and Franz-Josef-Land that observations show between 1999 and 2003 ($6.5*10^4$ km$^2$yr$^{-2}$; Kwok, 2009) would be slightly too small to be considered relevant for triggering a RICE.

*Surface heat fluxes:* Changes in atmospheric circulation and associated heat and moisture transport can also influence the

135 sea ice cover (Woods and Caballero, 2016; Boisvert et al., 2016). In support of this, our results show a negative trend in sea

Number: 1 Author: anonymous   Subject: Highlight   Date: 11/07/2022, 15:37:09
Looking at Figure 4, I see that the distances increase from (a) to (b) but in (c) only CESM-LE increases further. Perhaps rephrase and state "during the first half of the 21st century" ?

Number: 2 Author: anonymous   Subject: Highlight   Date: 11/07/2022, 15:38:01
Again, please specifiy briefly what choice of thresholds you tried.

Number: 3 Author: anonymous   Subject: Highlight   Date: 11/07/2022, 15:39:30
I don't understand how you obtained those % numbers -- did you calculate some sort of explained variance? How did you "related" the trends quantitatively? Deserves a couple of extra sentences.

Number: 4 Author: anonymous   Subject: Highlight   Date: 11/07/2022, 15:40:34
See my comment above -- explain a bit more why the approach of calculating OHT is physically meaningful or why the result is not sensitive to the choice of reference temperature.

Number: 5 Author: anonymous   Subject: Highlight   Date: 11/07/2022, 15:35:56
There are a lot of acronyms in this manuscrpt, which makes it difficult to understand for the reader who is not an expert on sea ice in CMIP6 models. Consider writing at least some of these acronyms out in full -- for example, "OHT", "ITE", "ITN", "SHF" could be used in figures (defined in corresponding figure caption) but written in full in the text.

[Figure]

[Figure]

**Figure 4.** The fraction of RICEs that show a simultaneous trend in the respective forcing parameter of more than one standard deviation. The drivers are BSO ocean heat transport (OHT), ice transport through the northern (ITN) and eastern gateway (ITE) and surface heat flux (SHF) in the southwestern Barents Sea (Figure 5c).

level pressure over the Fram Strait during ice loss events (Fig. 5a), associated with strengthening westerly winds over BSO and southerly winds over the central and northern Barents Sea. As a result, surface air temperatures increase in the northern Barents Sea (Fig. 5b) during ice loss. Warmer westerly winds also lead to reduced ocean heat loss in the ice-free southern Barents Sea, whereas more heat is lost in the northern Barents Sea as a result of more open ocean area (Fig. 5c; Skagseth et al., 2020). Considering surface heat fluxes in the permanently ice-free southwestern Barents Sea (16-38° E; 71-76° N) as a fingerprint of atmospheric forcing of ocean temperature and, hence, sea ice (Schlichtholz and Houssais, 2011), we find 65% (62%) of the RICEs to be associated with anomalous trends in SHF in CESM-LE (CESM-2C); decreasing (increasing) ocean heat loss corresponding to SIA decline (increase).

Number: 1 Author: anonymous   Subject: Highlight      Date: 11/07/2022, 15:40:53
See comment above.

Number: 2 Author: anonymous   Subject: Highlight      Date: 11/07/2022, 15:42:40
Does that mean you just counted the number of times that RICE were related to anomalous trends in SHF (i.e. where the trens in SHF went outside one standard deviation)? Is that the same way you get the percentages in Figure 4?

[Figure]

[Figure]

**Figure 5.** Linear trend of a) sea level pressure (SLP), b) surface air temperature (SAT) and c) surface heat flux (SHF) anomalies during ice loss events, averaged over all events. Ocean heat loss is defined as negative, meaning that positive (negative) anomalies refer to less (more) heat loss. The black box indicates the area for averaging SHF to assess its influence on RICEs. Crosses indicate areas where the trend is not statistically significant at the 95% confidence level.

Although the different drivers have been assessed and quantified individually, they are to some extent interconnected. For example, ice loss events are associated with an anomalous atmospheric circulation (Fig. 5a) that will influence both ocean heat transport (Herbaut et al., 2015), surface heat fluxes (Skagseth et al., 2020), and sea ice area transport (Kwok, 2009). 51 (50) % of the RICEs exhibit significant trends in both, OHT and SHF, emphasiszing their interconnection. A detailed analysis of these relationships is not presented here. However, removing (by regression) the linear signal associated with OHT from time series of regional winds over the Barents Sea suggests that atmospheric circulation (wind) anomalies are mainly effecting the sea ice cover through changes in BSO heat transport, consistent with the findings of e.g. Lien et al. (2017).

In general, we find that the relative importance of the different drivers varies somewhat during the simulations, but no systematic change is seen (Fig. 4). Differences between CESM-LE and CESM-2C are also small and show no consistency, suggesting that the forcing of RICEs remains unaffected by the underlying ice conditions.

**5 Rapid Sea Ice Changes in CMIP6 models**

The occurrence of rapid sea ice changes is further studied in a suite of CMIP6 models. Figure 6 shows the distribution of 5-year trends from the CMIP6 models under two emission scenarios. Most models show a distribution of trends that is fairly similar to, yet slightly narrower than the CESM. An exception is CanESM5 that simulates much weaker trends than the other models, likely as a result of the low average SIA in the model after the strong ice loss until 2030 (Fig. 6). During the 21st century SIA decreases in all model simulations and the distributions become more confined to weaker trends. Although the models agree on

Number: 1 Author: anonymous   Subject: Highlight     Date: 11/07/2022, 15:44:16

This would suggest that most of the BSO heat transports are volume-transport-driven, rather than driven by changes in temperature of the advected water. That devserves another sentence... Again, I refer to my comment on heat transports above.

Number: 2 Author: anonymous   Subject: Highlight     Date: 11/07/2022, 15:46:11

Do you mean that the sea ice loss occured only until 2030? Or does the low average SIA in the model occurs after a strong ice loss that alltogether occur before 2030? Please rephrase to clarify...

[revised manuscript text omitted]

Number: 1 Author: anonymous    Subject: Highlight    Date: 11/07/2022, 15:48:50

If sea ice change in the northern Barents Sea and the Eurasian basin near the continental slope is a result of Atlantification or, rather, a adriver of it may be an open question (or it's both) -- I'd rephrase this as something like "a visible change associated with future Atlantification".

Number: 2 Author: anonymous    Subject: Highlight    Date: 11/07/2022, 15:53:38

I would expect a proper citation of data sources here, i.e. cited here or in the main text and put in the reference list. At least the ERA5 and the SIC observations should be citable (with a doi) and available via a repository. Not sure about the model output. In fact, when following your link for NSIDC, there is a clear indication of "Citing these data": "Walsh, J. E., W. L. Chapman, F. Fetterer, and J. S. Stewart. 2019. *Gridded Monthly Sea Ice Extent and Concentration, 1850 Onward, Version 2*. [Indicate subset used]. Boulder, Colorado USA. NSIDC: National Snow and Ice Data Center. doi: https://doi.org/10.7265/jj4s-tq79. [Date Accessed]. "
Please check for the other data sources, as well.

---

## Author Response (AR2)

**List of relevant changes**

Based on the feedback of the reviewers, we have substantially revised and improved our manuscript. We have rewritten parts and changed the wording where necessary. We have also adapted the notation of units and numbers in accordance with the journal guidelines. Please find a list of all the significant changes below, and a detailed response to the individual reviewers on the following pages.

**Abstract**

- We have introduced the abbreviation "RICEs" and used it in the abstract.

**Introduction**

- Line 13: We have added references to observational studies about Arctic Amplification.

- Line 14: We now cite the IPCC-report following a suggestion from reviewer 1.

**Data and Methods**

- We have restructured the entire section following a suggestion from reviewer 2. We now first introduce the used data, including the different model, and its applicability to the region of interest. Then we explain more detailed the approach of dividing outputs from multi-member-ensemble simulations into internal variability and response to external forcing. Then we explain our methods, including the definition of RICEs and significant drivers of those. Finally we introduce the different variables used in the study.

- We have particularly clarified the idea of using large ensemble simulations.

- Line 69 and 74: We have specified different values that we have explored for the used thresholds.

- Line 74: We have clarified how we assess the relevance of the drivers of RICEs.

- Line 79: We have explained our calculation of OHT using a reference temperature and that our results are not sensitive to this exact choice.

**Sea Ice Loss and Variability in the Barents Sea**

- We have added colours to the notation of the sample size in Figure 2 following a comment from reviewer 1.

- In the caption of Figure 2 and in the following we are now more clear with our notation: "Externally-forced" refers to the ensemble mean state. We avoid confusing this with the "anthropogenic signal" as it is not entirely equivalent, and do not use that term in association with the simulations.

**Rapid Sea Ice Changes in CESM1**

- We removed the abbreviations OHT, SHF and BSO in the text following the suggestion of reviewer 1.

- We shortened the part about the displacement of the ice edge during RICEs.

- Introduced new subsection about the forcing of RICEs with bullet points addressing the individual variables following a suggestion of reviewer 2.

- Explanation about how to identify whether a driver is related to a RICE (previously line 115) is now moved to the Data and Methods section (now line 74).

- We state that we find no differences between ice growth and loss events before the individual sections for the drivers (now line 165). This sentence was previously at the end of this section.

- Added some additional info in caption of Figure 4.

- We no longer name the percentages for both CESM-LE and CESM-2C but only for CESM-LE as this makes it easier to read. The respective values for CESM-2C can be found in Figure 4.

- Line 192: Added a new study about the importance of atmospheric conditions for sea ice variability in the Barents Sea.

- Line 207: We have added a statement about volume transport, as suggested by reviewer 1.

- Line 214: Following a suggestion from reviewer 2, we have explored whether the strength of a RICE can be related to the linear combination of trends in the individual variables, but find no connection there. We now state this in the text and suggest the potential predictability by using more sophisticated approaches.

**Rapid Sea Ice Changes in CMIP6 models**

- We have removed the average SIC from the panels in Figure 6. Instead we have added the number of RICEs per ensemble member to the top left (SSP585) and right (SSP126) corner of the top panels. This is also indicated by the respective color.

- We have removed the number of RICEs and the ensemble-mean ice decline from the text to improve readability.

**Discussion and Conclusion**

- We are now consistently using the abbreviation "RICE" instead of spelling it out.

**Data Availability Statement**

- Added citations for the observational and reanalysis datasets. Citations for the model output can be found in section 2.

**Acknowledgements**

- We thank David Bonan for his valuable feedback on the manuscript.

**First Reviewer**

**Comment on egusphere-2022-324**
Anonymous Referee 1

Referee comment on "Rapid Sea Ice Changes in the Future Barents Sea" by Ole Rieke et al., EGUsphere, https://doi.org/10.5194/e 2022-324-RC1, 2022

I have been asked to review the manuscript "Rapid Sea ICe CHanges in the Future Barents Sea" by Rieke et al. The topic of the work is important for current climate resarch, as the Arctic is constantly changing alongside global warming. In particular, the Barents Sea is a part of the region representing an important connection between the Arctic basins and the remainder of the Eurasian shelves, and lower latitudes. Whereas the data basis for analysis appears state-of-the-art and the analytical approach overall sound there are several details that I suggest be clarified before publication. The method of relating different derived quantities to changes in sea-ice cover is not clearly explained. The calculation of heat transport has, in the past, been challenged, and at least a few sentences of explanation why the approach used here is valid would be in order. Thus, I suggest to publish this manuscript, subject to the modest corrections outlined in my comments. Note that I submit the comments as PDF comments in the form of a comment summary.

**We thank the reviewer for all the valuable feedback on our paper. We have thouroughly revisited our manuscript following the reviewer's comments and adjusted parts of the text and figures accordingly. The reviewer's main concern was the calculation of ocean heat transport. Ocean heat transport through individual sections (such as the BSO) must be calculated relative to a reference temperature (Tref), which is in principle arbitrary (Schauer and Beszczynska-Möller, 2009). "In our calculation of OHT, Tref = 0 °C. This is commonly used for OHT calculations in the Barents Sea (Årthun et al., 2012; Smedsrud et al., 2013; Koenigk and Brodeau, 2014; Li et al., 2017). We have calculated OHT using other choices of Tref and find that our results (specifically the magnitude of present and future OHT trends and their link to RICEs) are not sensitive to this (Fig. 1).**
**Please find below a detailed response to each comment.**

- Line 13: could also cite the current IPCC report: Meredith, M., M. Sommerkorn, S. Cassotta, C. Derksen, A. Ekaykin, A. Hollowed, G. Kofinas, A. Mackintosh, J. Melbourne-Thomas, M.M.C. Muelbert, G. Ottersen, H. Pritchard, and E.A.G. Schuur, 2019: Polar Regions. In: IPCC Special Report on the Ocean and Cryosphere in a Changing Climate [H.-O. Pörtner, D.C. Roberts, V. Masson-Delmotte, P. Zhai, M. Tignor, E. Poloczanska, K. Mintenbeck, A. Alegría, M. Nicolai, A. Okem, J. Petzold, B. Rama, N.M. Weyer (eds.)]. In press.)
  **We have included the citation of the IPCC report (new line 14).**

- Line 17: scenarios
  **Typo fixed (line 18.**

- Line 56: Please specify what reange of choices you explored (e.g. one standard deviation, three standard deviations...).
  **We have explored the dependency on our results based on very different thresholds, using criteria based on the internal distribution (1-2.5 standard deviations) and on external criteria (5-10 times the observed ice decline since 1979). A threshold lower than this does not justify the RICEs being "extreme" events and has thus not been explored. We now mention this in the text (line 69).**

- Line 56: .
  **Missing dot inserted.**

- Figure 1 Caption: (SIA).
  **Inserted**

- Line 61: This appears to be an arbitrary choice. Seawater at typical Barents Sea salinities in winter freezes around -2 deg C. Please elaborate a bit more how sensitive your results are to this choice of reference temperature. Refer to / cite the following publication: Schauer, U. and Beszczynska-Möller, A.: Problems with

[Figure]

Figure 1: Fraction of RICEs that can be related to trends in ocean heat transport using different reference temperatures. OHT based on the model output variable UET uses a reference temperature of $0\,°C$.

estimation and interpretation of oceanic heat transport – conceptual remarks for the case of Fram Strait in the Arctic Ocean, Ocean Sci., 5, 487–494, https://doi.org/10.5194/os-5-487-2009, 2009.
**As mentioned above, we have calculated OHT trends with different reference temperatures and find that our results are not sensitive to this choice (Fig. 1). The use of reference temperature is now discussed in the text, including a reference to Schauer and Beszczynska-Möller (2009) (line 80).**

- Figure 2: I know this may seem picky, but if you used the same colours for the nLE, n2C and nObs as in the legend it would be intuitively easier to associate the "n" to the correct set of data.
  **Good idea! We have changed the colours following the reviewer's suggestion.**

- Line 107: Looking at Figure 4, I see that the distances increase from (a) to (b) but in (c) only CESM-LE increases further. Perhaps rephrase and state "during the first half of the 21st century"?
  **We have removed this unclear statement.**

- Line 116: Again, please specifiy briefly what choice of thresholds you tried.
  **We have investigated thresholds ranging from one to three standard deviations. This does not qualitatively change the results in terms of the relative importance of the drivers, but only the absolute numbers. One standard deviation is sufficient to indicate a clear relation between the**

driver and the SIA trend, and provides large enough numbers to investigate the evolution of the drivers over time. This has been added to the text, now in the data and methods section (line 74).

- Line 118: I don't understand how you obtained those % numbers – did you calculate some sort of explained variance? How did you "related" the trends quantitatively? Deserves a couple of extra sentences.
  **These numbers simply describe the fraction of events where the respective trend of a driving mechanism exceeds our threshold of one standard deviation. We have clarified this in the text (line 74).**

- Line 121: See my comment above – explain a bit more why the approach of calculating OHT is physically meaningful or why the result is not sensitive to the choice of reference temperature.
  **See statement on OHT above.**

- Line 124: There are a lot of acronyms in this manuscrpt, which makes it difficult to understand for the reader who is not an expert on sea ice in CMIP6 models. Consider writing at least some of these acronyms out in full – for example, "OHT", "ITE", "ITN", "SHF" could be used in figures (defined in corresponding figure caption) but written in full in the text.
  **Following the reviewer's suggestion, we have removed some of the acronyms in the text. For example, acronyms of Barents Sea Opening, ocean heat transport and surface heat fluxes are now restricted to figures following the reviewer's suggestion.**

- Figure 4 caption: See comment above.
  **Acronyms now explained in figure captions.**

- Line 141: Does that mean you just counted the number of times that RICE were related to anomalous trends in SHF (i.e. where the trens in SHF went outside one standard deviation)? Is that the same way you get the percentages in Figure 4?
  **Yes, that is correct. See the reply above. We have clarified this in the text (line 74).**

- Line 149: This would suggest that most of the BSO heat transports are volume-transport-driven, rather than driven by changes in temperature of the advected water. That devserves another sentence... Again, I refer to my comment on heat transports above.
  **We find indeed that the wind influences the volume (and hence the heat) transport through BSO. This is consistent with previous studies that have found volume transport to be most important to heat transport variability on interannual to decadal time scales (e.g., Muilwijk et al., 2018; Årthun et al., 2019). This is now stated in the manuscript (line 157).**

- Line 158: Do you mean that the sea ice loss occured only until 2030? Or does the low average SIA in the model occurs after a strong ice loss that alltogether occur before 2030? Please rephrase to clarify...
  **We have clarified this in the text (line 173).**

- Line 193: See my comments on OHT above...
  **See statement on OHT above.**

- Line 208: If sea ice change in the northern Barents Sea and the Eurasian basin near the continental slope is a result of Atlantification or, rather, a adriver of it may be an open question (or it's both) – I'd rephrase this as something like "a visible change associated with future Atlantification".
  **We have rephrased this as suggested (line 219).**

- Data availability statement: I would expect a proper citation of data sources here, i.e. cited here or in the main text and put in the reference list. At least the ERA5 and the SIC observations should be citable (with a doi) and available via a repository. Not sure about the model output. In fact, when following your link for NSIDC, there is a clear indication of "Citing these data": " Walsh, J. E., W. L. Chapman, F. Fetterer, and J. S. Stewart. 2019. Gridded Monthly Sea Ice Extent and Concentration, 1850 Onward, Version 2. [Indicate subset used]. Boulder, Colorado USA. NSIDC: National Snow and Ice Data Center. doi: https://doi.org/10.7265/jj4s-tq79. [Date Accessed]. " Please check for the other data sources, as well.
  **We have included the citations as suggested for the observational dataset and the reanalysis**

data. Model output is available via the given links, citation of the models and simulations used in this study is provided in the data and methods section.

**References**

Koenigk, T. and Brodeau, L.: Ocean heat transport into the Arctic in the twentieth and twenty-first century in EC-Earth, Climate Dynamics, 42, 3101–3120, https://doi.org/https://doi.org/10.1007/s00382-013-1821-x, 2014.

Li, D., Zhang, R., and Knutson, T. R.: On the discrepancy between observed and CMIP5 multi-model simulated Barents Sea winter sea ice decline, Nature Communications, 8, https://doi.org/https://doi.org/10.1038/ncomms14991, 2017.

Muilwijk, M., Smedsrud, L. H., Ilicak, M., and Drange, H.: Atlantic Water Heat Transport Variability in the 20th Century Arctic Ocean From a Global Ocean Model and Observations, Journal of Geophysical Research: Oceans, 123, 8159–8179, https://doi.org/https://doi.org/10.1029/2018JC014327, 2018.

Schauer, U. and Beszczynska-Möller, A.: Problems with estimation and interpretation of oceanic heat transport – conceptual remarks for the case of Fram Strait in the Arctic Ocean, Ocean Science, 5, 487–494, https://doi.org/https://doi.org/10.5194/os-5-487-2009, 2009.

Smedsrud, L. H., Esau, I., Ingvaldsen, R. B., Eldevik, T., Haugan, P. M., Li, C., Lien, V. S., Olsen, A., Omar, A. M., Otterå, O. H., Risebrobakken, B., Sandø, A. B., Semenov, V. A., and Sorokina, S. A.: The Role of the Barents Sea in the Arctic Climate System, Reviews of Geophysics, 51, 415–449, https://doi.org/https://doi.org/10.1002/rog.20017, 2013.

Årthun, M., Eldevik, T., Smedsrud, L. H., Skagseth, Ø., and Ingvaldsen, R. B.: Quantifying the Influence of Atlantic Heat on Barents Sea Ice Variability and Retreat, Journal of Climate, 25, 4736–4743, https://doi.org/https://doi.org/10.1175/JCLI-D-11-00466.1, 2012.

Årthun, M., Eldevik, T., and Smedsrud, L. H.: The Role of Atlantic Heat Transport in Future Arctic Winter Sea Ice Loss, Journal of Climate, 32, 3327–3341, https://doi.org/https://doi.org/10.1175/JCLI-D-18-0750.1, 2019.

**Second Reviewer**

**Comment on egusphere-2022-324**
Anonymous Referee 2

Referee comment on "Rapid Sea Ice Changes in the Future Barents Sea" by Ole Rieke et al., EGUsphere, https://doi.org/10.5194/e
2022-324-RC2, 2022

This study investigates the trends of sea ice decline and variability, in the Barents Sea, as represented by climate models. More specifically, the study tries to relate the sea ice variability in the Barents Sea to its drivers, such as ocean or atmospheric forcing. The work is relevant, and deserves publication, but the manuscript needs to be clarified because it is difficult to follow the method or the conclusions. Basically, one gets the impression some important work has been done, but assembled in a messy way. If this manuscript is mostly the work of its first author, I would suggest the co-authors to review the manuscript as if they were reviewers, if it is a collective job then the manuscript could be reviewed by a colleague before submitting a revised version. I really think the work is interesting, but I recommend major revision because the manuscript needs clarification.

**We thank the reviewer for the valuable feedback! We have revisited the manuscript and changed text and figures accordingly. Following the reviewer's suggestions, we have in particular tried to improve the text. This includes a thorough review by all co-authors and an English-speaking colleague. We hope the reviewer finds our manuscript improved and easier to follow.**

**Please find below a detailed response to each comment.**

- Line 63 : F is not the advective heat flux per unit volume, but rather the advection of temperature per unit volume.
  **This is correct! We have changed this accordingly (line 78).**

- Line 57 : Can you rephrase this sentence, super heavy to read. You could reverse it. Trends in ocean heat transport via the Barents Sea Opening (BSO), sea ice area transport between Franz Josef Land and Novaya Zemlya (eastern gateway) and between Svalbard and Franz Josef Land (northern gateway), net surface heat fluxes, sea level pressure....
  **We have rewritten this sentence (line 82).**

- Section 2. Data and Methods :
  This section is very confusing. Each bit does have a meaning, but all the bits together are really hard to follow because you switch from one to the other. Could you split in subsections, or at least make paragraphs : basically it starts with the model references, then explains how you define RICEs, then the fluxes etc.
  **We have rearranged the paragraphs in the section following the reviewer's suggestion, as well as moved some text from the results section to the methods section. A description of the models used is now followed by the methodology. We believe that the section is now more clearly structured and easier to follow for the reader.**

- It is difficult to understand the concept of external forcing, can you explicitate as this idea comes back several times in the manuscript, there are references but it would be good to have a substantial description within the manuscript. Especially, how do you arrive at the conclusion that within the number of members taken, the mean represents only the external forcing ?
  **We have rewritten that part to be more specific about the approach. We now state the following in the manuscript following line 56:**
  **"Using multi-member ensemble experiments allows for a detailed investigation of internal variability. The setup of the individual simulations differs only in slightly perturbed initial atmospheric conditions. Since the external forcing is the same for each simulation, the differences between the individual simulations are thus solely a result of internally-generated variability (Deser et al., 2020). The externally-forced contribution of e.g., sea ice change, is thus defined as the ensemble mean change (either from the 40 members of the CESM-LE or each CMIP6 model). To isolate the internal variability, we subtract the ensemble mean from each ensemble member."**

**This approach is one of the major advantages of large ensemble simulations and has been used in several studies before (Deser et al., 2012, 2014, 2020; Bonan et al., 2021).**

- In the middle of the section you already refer to Fig.2 in which the distribution of something obviously is plotted, but the reader does not even know what (although obvious for you). The beginning of the section describes models used, then you switch to methods to estimate events, and then at the end you switch back to model qualities... such a mess, please rewrite the entire section to split ideas so that the reader can follow. **As mentioned above we have rearranged this section. The reference to Figure 2 has been removed.**

- Section 4:
  Could you explain why the mean internal trend is not zero in Figure 2, this sounds strange as you mention these are only deviations from the ensemble mean, which suggests it is substracted. This appears to be confusing even further later in Line 92.
  **The black line in Figure 2 does not represent the mean of the internally-driven sea ice trends, but rather the externally-forced sea ice loss in CESM-LE. The latter is, as explained above, based on the ensemble mean sea ice concentration, i.e., the thick orange line in Figure 1b. We have changed the notation to avoid confusion.**

- The distribution of observed trends after 1920 is shown for comparison, and is similar to both, the historical simulations of CESM-LE (not shown) and the future simulations between 2007 and 2025 (Fig. 2a). I guess you refer to the Observations in Fig. 2a, make it clear by using similar words.
  **Yes, this is correct. We have modified this in the text (line 100).**

- Line 115 : you switch here to the main interest of the paper, the underlying processes behind the RICEs. I would create another section, or a subsection. **Good idea! We have added a new subsection as suggested.**

- Line 120 : heavy notation since you use twice semi-columns, I would suggest to switch to a bullet list or use sub-sub-sections. Add for each forcing its abbreviation so one identifies it when reading Figure 4 without reading the caption.
  **We have added bullet points with abbreviations as suggested.**

- A general comment about the method, the relation with the underlying processes is presented is only presented here as a kind of correlation. I think it would give more added value to your work if you could show you can actually reconstruct the signal of RICEs through an empirical function, it is just a recommendation but i do not think it is a lot of work to do. This would permit to dis-entangle processes which are, as you suggest, counted twice as for example a strong temperature anomaly in the atmosphere can also be related with a stronger heat transport through the BSO. You could show that you can reconstruct the signal of RICEs with the same accuracy or almost by eliminating one of the processes.
  **This is a good suggestion and we agree that this would add value to our analysis. However, although most RICEs are clearly related to trends in the chosen drivers, their strength does not show a strong relationship (in a way that particularly strong RICEs would be related to stronger trends in e.g. OHT). Thus, reconstructing the magnitude of RICEs using multiple linear regression did not prove to be successful. We have added this result to the manuscript (line 162). We note that our method of relating a fraction of RICEs to specific drivers are in line with the approach of Auclair and Tremblay (2018) who studied rapid ice loss events (in summer) and its relation to OHT.**

- Section 5 and Conclusions : Too many number to read in Section 5, it's difficult to follow. Can you sustain a bit further in terms of processes all the differences between models ?
  **To improve readability, we have reduced the amount of numbers in the text. The number of RICEs is, for example, now rather displayed in Figure 6.**
  **As stated in the manuscript, we analyse RICEs in five CMIP6 models in order to assess whether the number of RICEs in CESM, and their future change, is representative. To understand the differences between the models, in terms of their mean sea ice state, their variability, and their**

     **sensitivity to forcing, would require a detailed mechanistic analysis that is beyond the scope of this study.**

- I'll coment further on this part on the revised version.

- 7.7 $10^4$ km2yr1 and such numbers : Do not use the * sign, and put some spacing. Check in other articles of the same journal on how such number are written.
  **We have changed the notation of this numbers and now use spacing between and before units.**

- dd some dots in some places before beginning a new phrase: based on Walsh et al., 2017). Please check sentences in general, and avoid long sentences.
  **This has been fixed.**

- Check the case according to journal requirements, I suspect eastern gateway should be written Eastern gateway (this is just an example, directions in general should be writen East, North, etc. I believe).
  **The English language guidelines of the journal recommend to write east, northern, etc. using small letters unless part of a name ("North America") or established expression. Our writing follows these guidelines.**

**References**

Auclair, G. and Tremblay, L. B.: The Role of Ocean Heat Transport in Rapid Sea Ice Declines in the Community Earth System Model Large Ensemble, Journal of Geophysical Research: Oceans, 123, 8941–8957, https://doi.org/ https://doi.org/10.1029/2018JC014525, 2018.

Bonan, D. B., Lehner, F., and Holland, M. M.: Partitioning uncertainty in projections of Arctic sea ice, Environmental Research Letters, 16, https://doi.org/https://doi.org/10.1088/1748-9326/abe0ec, 2021.

Deser, C., Phillips, A., Bourdette, V., and Teng, H.: Uncertainty in climate change projections: the role of internal variability, Climate Dynamics, 38, 527–546, https://doi.org/https://doi.org/10.1007/s00382-010-0977-x, 2012.

Deser, C., Phillips, A. S., Alexander, M. A., and Smoliak, B. V.: Projecting North American Climate over the Next 50 Years: Uncertainty due to Internal Variability, Journal of Climate, 27, 2271 – 2296, https://doi.org/ 10.1175/JCLI-D-13-00451.1, 2014.

Deser, C., Lehner, F., Rodgers, K. B., Ault, T., Delworth, T. L., DiNezio, P. N., Fiore, A., Frankignoul, C., Fyfe, J. C., Horton, D. E., Kay, J. E., Knutti, R., Lovenduski, N. S., Marotzke, J., McKinnon, K. A., Minobe, S., Randerson, J., Screen, J. A., Simpson, I. R., and Ting, M.: Insights from Earth system model initial-condition large ensembles and future prospects, Nature Climate Change, 10, 277–286, https://doi.org/ https://doi.org/10.1038/s41558-020-0731-2, 2020.

**Second review**

**Comment on egusphere-2022-324**
 Anonymous Referee 1

Referee comment on "Rapid Sea Ice Changes in the Future Barents Sea" by Ole Rieke et al., EGUsphere, https://doi.org/10.5194/e
2022-324-RC1, 2022

I suggest very minor corrections to this significantly improved / revised manuscript. Abbreviations are not, yet, consistently defined and introduced; for example, OHT / ocean-heat transport is introduced, then abbreviation used, then spelled out in full, then again introduced... As for the heat transport I find the treatment of the reference temperature a bit thin. At least the authors could warrant another sentence, for example, also citing some of all of https://tos.org/oceanography/article/arctic-ocean-boundary-exchanges-a-review
(her you can find further references regarding heat transport calculation and reference salinity)
https://www.nature.com/articles/s41558-020-00941-3
https://journals.ametsoc.org/view/journals/phoc/48/9/jpo-d-17-0239.1.xml

**We thank the reviewer for the suggestions. We have checked the definition of abbreviations and are now more consistent with this. We have added another sentence and reference on the calculation of ocean heat transport.**

Anonymous Referee 2
Referee comment on "Rapid Sea Ice Changes in the Future Barents Sea" by Ole Rieke et al., EGUsphere, https://doi.org/10.5194/e
2022-324-RC1, 2022

This is my second review of this manuscript. I think the second version has greatly improved in readibility. The authors have provided a very detailed response to my comments, which have been for most cases taken into account, and the research work is relevant so I think this manuscript is now mature for publication. I regret however that there is not a deeper analysis of processes as I had suggested in my first review, but I think it would have given a stronger substance to the paper.

**We thank the reviewer, and are happy to hear that the manuscript has improved. Unfortunately, a deeper analysis of the processes would go beyond the scope of this work.**